# Fat tails and the need to disclose distribution parameters of qEEG databases

**Guilherme Wood** [1]*, **Klaus Willmes**[2], **Jan Willem Koten**[1], **Silvia Erika Kober**[1]

**1** Institute of Psychology, University of Graz, Graz, Austria, **2** Neurological Clinic—Neuropsychology, RWTH Aachen, Aachen, Germany

* guilherme.wood@uni-graz.at

## Abstract

Neurometry (a.k.a. quantitative EEG or qEEG) is a popular method to assess clinically relevant abnormalities in the electroencephalogram. Neurometry is based on norm values for the distribution of specific EEG parameters and believed to show good psychometric properties such as test-retest reliability. Many psychometric properties only hold under the Gaussian distribution and become problematic when distributions are fat-tailed. EEG signals are typically fat-tailed and do not show fast convergence to a Gaussian distribution. To circumvent this property of EEG, log-transformations have frequently, but not always been employed. In Monte Carlo simulations, we investigated the impact of fat-tails (i.e. deviations from Gaussian) on the cut-off criteria and changeability of what in neurometry is termed "abnormal EEG". Even slight deviations from the Gaussian distribution as measured by skewness and kurtosis lead to large inflation in the number of false positive qEEG findings. The more stringent the cutoff value adopted, the larger the inflation. Moreover, "abnormal EEG" seems to recover spontaneously at rates not compatible with the alleged test-retest reliability of qEEG. Alternative methods should be employed to determine cut-off values for diagnostics purposes, since a large number of false positive results emerge even when slight deviations from the Gaussian distribution are present. We argue that distribution properties of qEEG databases should be disclosed in much more detail by commercial providers to avoid questionable research practices and promote diagnostic transparency. We provide recommendations for the improvement of psychometric properties of existing qEEG databases.

## Introduction

Neurometry (a.k.a. quantitative EEG or qEEG) is the popular name given to the adaptation of classic psychometric measurement models to the evaluation of EEG data [1]. Typically, EEG parameters are measured in a "large" sample and norm values are calculated and employed as a reference to classify EEG signals as normal or abnormal and suggest specific forms of "cure" to "normalize" EEG classified as deviant. More or less extreme cut-off values are used to classify the EEG of individuals as "normal" and "abnormal" [2] and in case of abnormality, an intervention such as neurofeedback [3] can be applied to "normalize" the EEG. Several parameters of EEG have been treated in this way: absolute and relative frequency power, coherence,

**Data Availability Statement:** This is a mathematical simulation. The script for the calculations is attached. Each user can generate the data anew with ease.

**Funding:** The publication costs have been funded by the University of Graz.

**Competing interests:** The authors have no conflicts of interest to declare.

phase, and event-related potentials [4]. A PubMed search for the term "qEEG" alone yields to the present time point over 1000 entries.

The rationale behind the construction of normative databases is strongly dependent on assumptions regarding the distribution of EEG population parameters. Ideally, these parameters converge quickly to a Gaussian distribution and the true population values for mean and variance of the EEG can be estimated with reasonable effort. Results acceptable for diagnostic purposes can be achieved for the mean with sample sizes of n = 30 and the variance with n = 750 [5]. This is the *n* when considering a reasonably useful estimate that will not deviate more than d = 5% from the correct value with a confidence of 95%. Taking a less stringent accuracy criterion may not be problematic when evaluating estimates close to the mean, but problematic for more extreme observations. While an error of 10% is still fine when observations are 1σ away from the mean, an error of 10% when estimating an observation 2σ away from the mean is much more pronounced. An estimate of the variance is paramount for the development of norms, since the standard deviation serves as the unity to describe how far an observation is from the mean. The proportion of observations more extreme than a z-score of +/- 2 or 3 is given by the density function of the Gaussian distribution (i.e. ≅ 5% and 0.1%, respectively) and serves well as a cut-off value for diagnostics. An inaccurate estimate of the variance leads to imprecise z-scores and cut-off values that are useless for individual diagnostics because the proportions of values exceeding the cut-off values may be much higher or lower than expected. Test-retest reliability measures provide the basis for measurement of intervention effects, since they estimate the expected changes in test scores which are due to measurement error alone. One important caveat for the usefulness of test-retest reliability is the homoscedasticity of measurement error. When skewness and kurtosis deviate from a Gaussian distribution, measurement error becomes more heteroscedastic because of one or both of the distribution's tails, and test-retest reliability estimates become invalid for observations far from the mean. In the present study, we will investigate the impact of deviations from a Gaussian distribution to the practice of comparing EEG with EEG norms using cut-off values.

Authors active in the development of neurometric tools tend to underscore that EEG parameters such as absolute power, relative power, phase, and coherence follow a Gaussian distribution. As reported for instance by [6, 7] (Table 1, p. 101), skewness and kurtosis deviate from normality to a certain degree in qEEG databases, but for decades this has been treated as negligible, but a psychometric proof of this claim was never presented. To the contrary, publications on qEEG databases rarely present information on the higher moments of the distribution of EEG parameters. This is highly problematic, since EEG data can show fat-tailed distributions such as a lognormal distribution [8] and require the use of mathematical transformations to achieve a Gaussian distribution.

The lognormal distribution can behave differently according to its standard deviation [9]. The lognormal distribution with a small standard deviation is almost symmetric around the mean and behaves indeed very similarly to the Gaussian, but starts very quickly to behave as a fat-tailed distribution, when its variance increases. When the distributions are truly lognormal, the logarithm of raw values behaves as a Gaussian and the assumptions for diagnostics are met. Obviously, transformations have to be chosen taking into consideration the properties of data at hand and only wrong transformations can damage data properties. Crucially, there is evidence from Box-Cox investigations, that EEG parameters can be distributed in a more extreme way than lognormal [10] (Table 2, p. 210). In such cases, a log-transformation is unable to bend the data distribution to a Gaussian and many of the properties of fat-tailed distributions will remain in the data even after a log-transformation and several of the measurement assumptions cannot be met for any practical purpose.

One characteristic of fat-tailed distributions is the higher probability density of observations far from the mean in comparison to the Gaussian distribution. The more fat-tailed a distribution is, the more observations are farther than 2, 3, 4 standard deviations from the mean, rendering these values useless as cut-offs (Fig 1). For instance, under a fat-tailed log-normal distribution with mean 0 and standard deviation 1, many more observations will be more extreme than 2 or 3 standard deviations than in a Gaussian distribution (i.e. 25% and 14%, respectively, and not only 5% and 0.1%, as under a Gaussian). These values illustrate how large the inflation in the number of false positives is when taking $2\sigma$ (0.25/0.05 = 500%) or $3\sigma$ (0.14/0.001 = 14000%) as cut-off values for neurometrics. When considering cut-off values based on the standard deviation, fat-tailed distributions lead to the inflation of occurrence of observations more extreme than the cutoff.

Moreover, estimating the moments of fat-tailed distributions (mean, standard deviation, skewness and kurtosis) is not an easy task because of slow convergence. Even after collecting many thousands of observations, sample estimates can be far away from the true population value. While one can rely on $n = 30$ to provide a not too bad estimation of the population mean of a Gaussian distribution, some common fat-tailed distributions would need no less than $10^9$ observations for estimation of the mean [9]. Even if many fat-tailed distributions converge to a Gaussian distribution when n approaches infinity, slow convergence implies sample sizes prohibitively large by many orders of magnitude in comparison to the Gaussian. For fat-tailed distributions, the empirical distribution does not reflect the true statistical properties of the population, since typical sample sizes of hundreds or even some thousands of participants are too small to allow useful estimation. This is problematic especially at the extremes of the distribution, which are central for diagnostics of "abnormal" EEG.

Fat-tailedness also interferes with the computation of score differences, which are essential when comparing qEEG scores obtained before and after an intervention. Due to an intervention, neurometrics specialists aim at a "normalization" of the EEG, which is expressed by a correction of extreme scores obtained prior to a treatment to levels typically observed in the population. Database providers use to report good or even excellent test-reliability scores for qEEG parameters [11–13]. Thanks to the test-retest reliability of qEEG databases, one can compute the critical difference of qEEG scores, which is the smallest non-trivial score difference, i.e. a score too large to be attributed to chance alone. In this sense, the critical difference helps accounting for the effects of regression to the mean and can therefore be taken as genuine evidence of the effectivity of a therapeutic intervention and is a great tool to evaluate the effects of heteroscedasticity on diagnostics and evaluation practices typical in neurometry. Under the Gaussian distribution, individual measurement error (i.e. $SE = z_\alpha \sqrt{1 - \rho}$ for

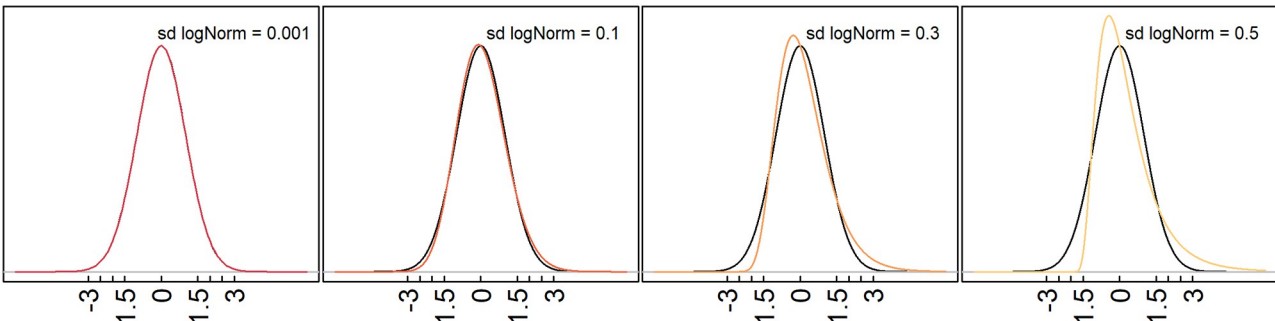

**Fig 1. Comparison of the densities of the Gaussian distribution (black line) and lognormal distributions, the latter with increasing variances.**
Note that at very small variance values the lognormal is indistinguishable from the Gaussian, but small increases in variance lead to large deviations at the tails.

z-transformed variables with σ = 1) is homoscedastic, so that the probability of test scores to surpass the critical difference by chance is the same for any test score regardless of how far it is from the mean. This is not true, however, when distributions are fat-tailed. In case of variables with excess skewness and kurtosis, heteroscedasticity generates larger critical differences far away from the mean than closer to it and give rise to the artificial impression of large improvements where there is none.

In the following, we will employ descriptions of qEEG distributions from the literature and test their susceptibility for inflation of positives and tail heteroscedasticity using Monte Carlo simulations.

## Materials and methods

The degree of deviation from a Gaussian distribution observed in popular qEEG databases can be estimated using the empirical values of variances, skewness, and kurtosis reported in the literature (e.g. [6], Table 2.2, p. 16 and [7], Table 1, p. 101). Thatcher (2010, Table IV, p 141) presents even higher skewness and kurtosis values for a LORETA normative database. These authors published skewness and kurtosis of their database as percentage value in relation to the moments of a Gaussian. Inspection of some of these values reveal deviations from Gaussian in the order of 0% to 50%. Although it is not explicitly reported in these publications, we assume that the sample moments are higher than the Gaussian because of the well-known fat-tail properties of EEG signals [8, 10]. This means that simulations with kurtosis values covering the interval between 3 and 4.5 are required to understand the diagnostic properties of qEEG data, considering that the Gaussian distribution has a kurtosis of 3.

### Monte Carlo simulation of cut-off values

We investigated the proportion of observations more extreme than typical cut-off values used to classify qEEG as normal or abnormal. Typical cut-off values of 2σ and 3σ as well as cut-off values of 1.5σ and 2.5σ were included in the simulations to improve the visualization of results. Nine different values ranging from 0.001 to 0.5 were used as the standard deviation to generate data with a lognormal distribution. Data drawn from a lognormal distribution with a very small value of sigma yields moments very close to the normal distribution ($skewness_{lognormal}$ σ = 0.001, $kurtosis_{lognormal}$ σ ≅ 3) while those of a lognormal distribution with a larger standard deviation = 1 are much higher (skewness = 5.75, kurtosis = 80.5) than those of a Gaussian. Simulated data were z-standardized to have a mean value of 0 and a standard deviation of 1 to allow the calculation of the cut-off values of 1.5, 2, 2.5 and 3σ and the comparison with the Gaussian distribution. Comparisons consisted of counting the number of values more extreme than the cut-off value in the lognormal and in the Gaussian distribution and dividing the first by the second number. When both values are comparable, this index is approximately one. When the lognormal distribution generates more positives, the value gets bigger than one and indicates an inflation of positives in comparison to the Gaussian. Finally, when the lognormal distribution generates less positives, the value gets smaller than one and indicates a deflation of positives in comparison to the Gaussian. In total, 36 conditions (4 cut-off values * 9 variance levels) were simulated with k = 10000 repetitions in each cell. In each repetition, one comparison of lognormal and Gaussian data was performed. Sample size was set at n = 1000 observations to reproduce the size of a large qEEG database. In the Supporting Information file, simulations of the inflation of false-positive values observed under other fat-tailed distributions ($Chi^2$-square, T distributions) are presented.

Monte Carlo simulation of score changes and test-retest estimates: Aim of this simulation is to understand the effect of non-normality on the interpretability of test score differences.

Usually, clinically relevant improvements in test-scores are attributed to the effectiveness of the intervention, but this must not be the case. For each test score difference there is a probability p of a false positive, i.e. test-scores show an improvement, which is due merely to random numeric fluctuation. One way to evaluate improvements in performance due to an intervention is the computation of the so-called critical difference in test scores [14]. The critical difference describes the minimal size of test score difference that is associated with a (low) probability p = α of being produced by measurement error alone. Formula (1) illustrates the calculation of the critical differences ([15], adapted from formula 7.2.10, for z-transformed variables with σ = 1), where Y represents the qEEG scores typically obtained before and after an intervention. As can be seen in (1), the critical difference is inversely related to the test-retest reliability $\rho_{jj}$ of test-scores. The larger the test-retest reliability of test scores, the smaller the smallest meaningful difference score.

$$crit\left(Y_{pre} - Y_{post}\right) = z_{1-\alpha/2}\sqrt{2(1 - \rho_{jj})} \tag{1}$$

QEEG parameters typically are reported to show moderate to high test-retest reliabilities [11–13]. Under homoscedasticity, the size of observed differences between test scores is independent of the absolute value of the scores. For a Gaussian distribution, the size of test score differences observed for each test score is comparable and the test-retest reliability can be used to determine an upper bound for the size of test score differences that is caused by random fluctuation alone. This value is valid in the whole range of possible test score values. Under heteroscedasticity, this is not the case. We generated pairs of variables with mean = 0, variance = 1, and with test-retest correlation $\rho_{jj}$. We then established the critical difference given the size of the α-level and the test-retest correlation $\rho_{jj}$.

Correlations of the same size as the test-retest reliability estimates published in the literature were employed to calculate the critical difference. These variables were drawn either from a Gaussian distribution or from nine different log-normal distributions with the same degree of logarithmic compression as those employed in the simulation of cut-off values (see above). Four different test-retest correlation values were employed ($\rho_{jj}$ was set to 0.5 (poor reliability), 0.7 (acceptable reliability), 0.8 (good reliability), and 0.95 (excellent reliability)) and 9 degrees of logarithmic compression (see Table 1). The number of repetitions for each experimental cell in the design was k = 10000. The first variable in each pair represented the pre-test score and the second the post-test score. The difference $D_{Ypre-Ypost}$ was calculated. The presence of heteroscedasticity was evaluated applying the Breusch-Pagan test [16, 17] to a linear model relating $D_{pre-post}$ to the pre-test score. The rationale is that under the Gaussian distribution, the model should have no predictive power and yield a beta coefficient of 0; the residuals of this model will be strictly homoscedastic. Under logarithmic compression, the regression coefficient will be different from 0 and the residuals will assume increasing levels of heteroscedasticity. Moreover, under an excess of skewness and kurtosis, the number of difference scores fulfilling the condition $D_{pre-post} > crit\left(Y_{pre} - Y_{post}\right)$ should increase, particularly far from the distribution mean.

**Table 1. Skewness and kurtosis of artificial data reproducing the typical properties of qEEG data.**

| σ of EEG distribution | 0.001 | 0.100 | 0.157 | 0.214 | 0.271 | 0.329 | 0.386 | 0.443 | 0.500 |
|---|---|---|---|---|---|---|---|---|---|
| **Skewness** | 0 | 0.30 | 0.48 | 0.65 | 0.84 | 1.00 | 1.24 | 1.47 | 1.71 |
| **Kurtosis\*** | 1 | 1.04 | 1.11 | 1.22 | 1.37 | 1.57 | 1.82 | 2.15 | 2.55 |

*Kurtosis is expressed as kurtosis(lognormal)/kurtosis(Gaussian)

## Statistical analysis

Simulations were programmed in R [18] using the packages base, stats v4.1.0, moments v.014 and lmtest v. 0.9–39. The scripts employed to generate data are available in the S1 Data. In simulation 1, probabilities of reaching a value higher than the cut-off were calculated using the cumulative density function of the normal distribution and were compared with the empirical probability values obtained for the different instances of the lognormal distribution.

## Results

Monte Carlo simulation of cut-off values: The proportion of values exceeding the cut-off were obtained by thresholding log-normally distributed data with different variances using cutoff values between 1.5 and 3 standard deviations (Table 1 and Fig 2). As can be seen in Fig 2, none of the cut-off values leads to a number of positives comparable to that of the Gaussian. Results

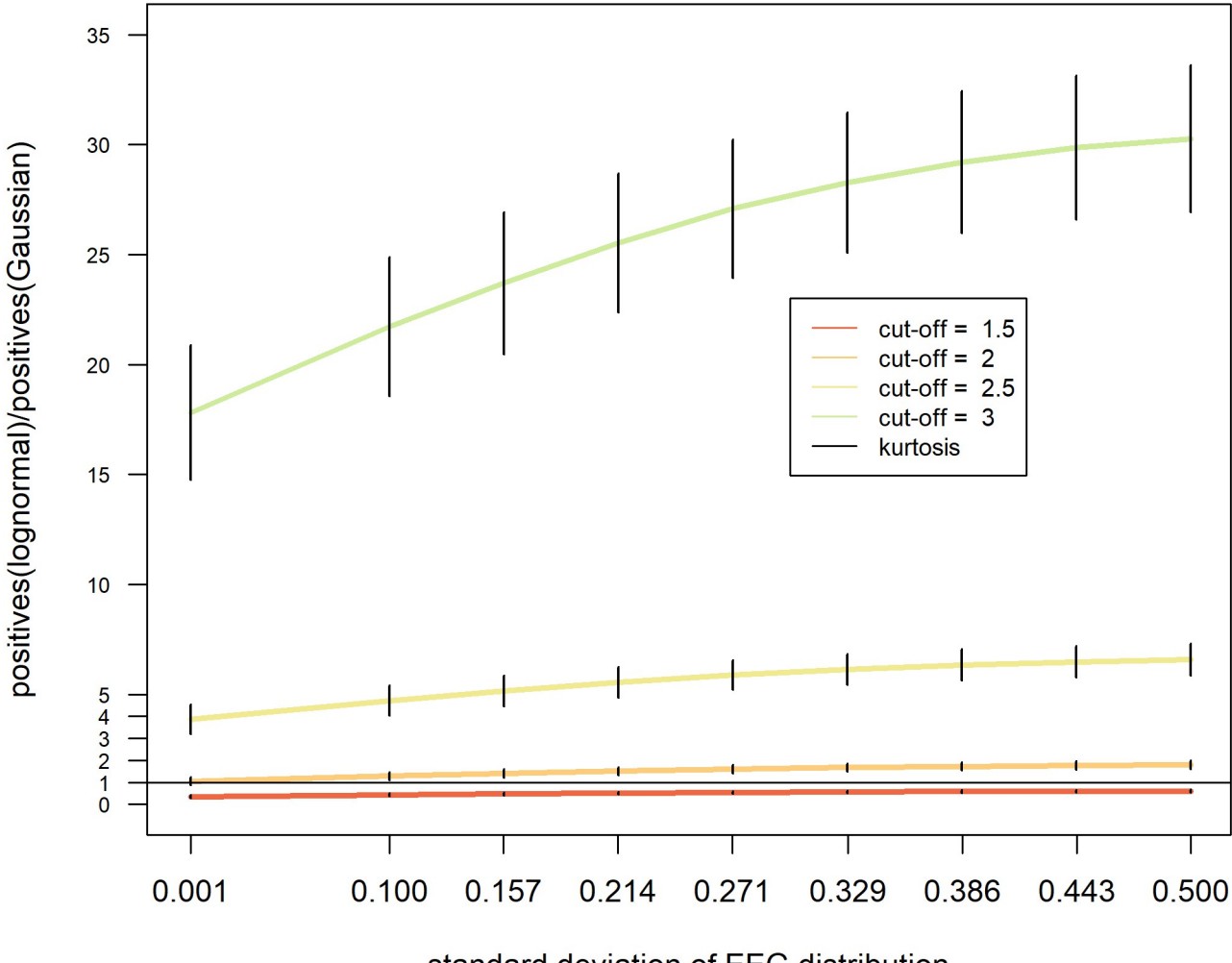

**Fig 2. Inflation of the number of positive observations more extreme than cut-off values.** The y-axis represents the proportion of values in EEG data exceeding the cut-off value divided by the expected frequency of a Gaussian distribution. In this simulation the cut-off values were set at 1.5, 2, 2.5 and 3σ.

of Simulation 1 show clearly how badly one can misinterpret the diagnostic value of EEG parameters, when these data deviate from a Gaussian distribution. A mismatch between the Gaussian and the lognormal distribution manifests itself in different directions and intensities depending on the cut-off value employed. On the one hand, liberal cut-off values (cut-off = 1.5) lead to an underestimation of the number of positive cases, which is worse when the kurtosis of the lognormal distribution is close to that of the Gaussian (σ of the EEG distribution = 0.1). Under these circumstances, the number of positives detected ranges exceeds the Gaussian from 9% to 167%. On the other hand, more conservative cut-offs produce a gross overestimation of positives, ranging between 16% and 1156%. Even in the case of a cut-off = 2σ, which generates the mildest misestimations, an inflation of 4% in the Kurtosis leads to an inflation of 32% in the number of positives. The size of the inflation increases rapidly and reaches 67% even when the kurtosis of the lognormal distribution is only about 20% higher than that of a Gaussian.

Fig 3 depicts the deviations from a Gaussian distribution observed when increasing the logarithmic compression of the data. It is evident that the extremes of the distribution deviate the most and are therefore more affected by the properties of the distribution.

Monte Carlo simulation of score changes and test-retest estimates: Table 2 depicts the average p-values obtained when testing for heteroscedasticity with the Breusch-Pagan test. An excess of kurtosis of 15% to 20% leads to systematic heteroscedasticity regardless of the level of test-retest reliability. Interestingly, the lowest levels of heteroscedasticity were observed when test-retest reliability is very high ($\rho_{jj}$ = 0.95). Still, even under these optimal test-retest reliability conditions significant heteroscedasticity is observed as an excess of kurtosis reaches 27%. Given these results, it is informative to compare the slope of the linear regression of $D_{pre-post}$ on the pre-test score $Y_{pre}$. As expected, this slope is 0 under the Gaussian distribution. Table 2 presents the slope values as $\log_{10}(1 + slope)$. When using this scaling, values larger than 0 in Table 2 reveal a positive correlation between the pre-test scores and the critical differences. The larger the pre-test score, the larger are the ($Y_{pre} - Y_{post}$) differences. Values in Table 2 increase for all test-retest reliability levels depending on the kurtosis excess. Finally, we computed the number of cases in which $D_{pre-post} > crit$ ($Y_{pre} - Y_{post}$) and the pre-test scores were larger than 2σ. Table 2 shows that the stronger the kurtosis excess, the larger the proportion of values $D_{pre-post} > crit$ ($Y_{pre} - Y_{post}$). Numerical values depict the multiplicative factor related to each cell in the design. The value of 1 indicates that the number of cases is comparable to that obtained under Gaussian distribution. As can be observed, the proportion of values exceeding the critical difference is larger than under the Gaussian for each level of logarithmic compression. More importantly, the higher the test-retest reliability, the larger the number of observations with high levels at pre-test and a large pre-post difference (i.e. a case showing what neurometricians call "normalization of EEG"). The values presented in Table 2 show that an inflation of more than 30% is observed with very small kurtosis excess and this inflation increases up to 370% in case of larger kurtosis excess and excellent test-retest reliability.

## Discussion/Conclusion

In the present study, we investigated how fat-tails of EEG data impact diagnostic criteria based on the assumption of a Gaussian distribution. Two main results emerged from our simulations. First, usual cut-off values based on standard deviations lead to a dramatic inflation in the number of false-positives even when deviations from the Gaussian distribution are quite modest. Second, the same modest deviations from the Gaussian distribution can lead to strong estimation bias in the extremes of the distribution, which leads to heavy misestimation of individual scores. In the following we will discuss these results in more detail.

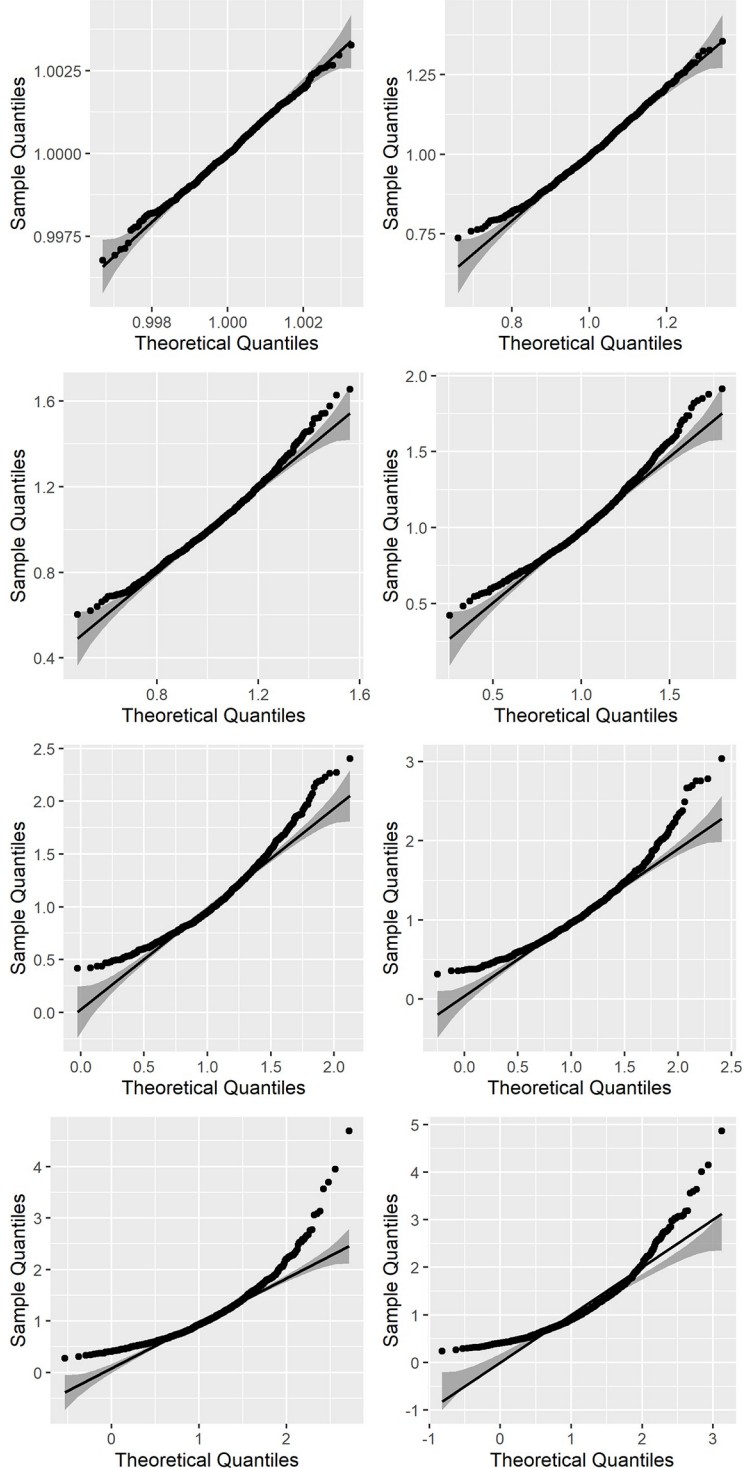

**Fig 3. The curvature in the normal Q-Q plot highlights the disagreement between the simulated data and the Gaussian model.**

**Table 2. Heteroscedasticity, dependency on initial scores, and inflation of false positives as a function of the standard deviation of log-normal distribution used in the simulations.**

| SD of EEG distribution | Breusch-Pagan test | | | | | | | | |
|---|---|---|---|---|---|---|---|---|---|
| | **0.001** | **0.100** | **0.157** | **0.214** | **0.271** | **0.329** | **0.386** | **0.443** | **0.500** |
| $\rho_{jj} = 0.5$ | -0.43 | -1.13 | -1.97 | -3.02 | -4.26 | -5.49 | -6.82 | -8.06 | -9.23 |
| $\rho_{jj} = 0.7$ | -0.43 | -1.26 | -2.27 | -3.63 | -5.08 | -6.72 | -8.33 | -9.96 | -11.63 |
| $\rho_{jj} = 0.8$ | -0.43 | -1.15 | -1.99 | -3.11 | -4.34 | -5.67 | -7.02 | -8.48 | -9.60 |
| $\rho_{jj} = 0.95$ | -0.44 | -0.63 | -0.89 | -1.23 | -1.58 | -1.94 | -2.34 | -2.69 | -3.03 |
| | Slope of the difference $D_{pre\text{-}post}$ on the pre-test scores $Y_{pre}$ | | | | | | | | |
| | **0.001** | **0.100** | **0.157** | **0.214** | **0.271** | **0.329** | **0.386** | **0.443** | **0.500** |
| $\rho_{jj} = 0.5$ | 0 | 0.02 | 0.03 | 0.04 | 0.05 | 0.06 | 0.06 | 0.07 | 0.08 |
| $\rho_{jj} = 0.7$ | 0 | 0.01 | 0.02 | 0.03 | 0.04 | 0.04 | 0.05 | 0.06 | 0.06 |
| $\rho_{jj} = 0.8$ | 0 | 0.01 | 0.02 | 0.02 | 0.03 | 0.04 | 0.04 | 0.05 | 0.05 |
| $\rho_{jj} = 0.95$ | 0 | 0.01 | 0.01 | 0.01 | 0.01 | 0.02 | 0.02 | 0.02 | 0.03 |
| | (Values exceeding the critical difference / Gaussian) -1 | | | | | | | | |
| | **0.001** | **0.100** | **0.157** | **0.214** | **0.271** | **0.329** | **0.386** | **0.443** | **0.500** |
| $\rho_{jj} = 0.5$ | 0 | 0.01 | 0.32 | 0.15 | 0.27 | 0.34 | 0.32 | 0.41 | 0.30 |
| $\rho_{jj} = 0.7$ | 0 | 0.23 | 0.55 | 0.66 | 0.69 | 0.69 | 0.61 | 0.96 | 0.93 |
| $\rho_{jj} = 0.8$ | 0 | 0.36 | 0.35 | 0.49 | 0.68 | 0.89 | 0.86 | 0.87 | 1.03 |
| $\rho_{jj} = 0.95$ | 0 | 0.24 | 0.53 | 0.49 | 0.54 | 0.75 | 0.69 | 0.92 | 1.07 |

The outcomes of Simulation 1 provide clear evidence that the diagnostic value of qEEG parameters can be badly overestimated, when these data deviate from a Gaussian distribution. When the cut-off is 1.5σ, the lognormal produces considerably less positives than the Gaussian, while cut-off values of 2.5 σ and 3σ produce dramatic inflation in the number of positives. In case of 2σ, the number of positives is close to the expected when the lognormal distribution resembles the Gaussian the most (i.e. 32% excess of positives), but it starts to deviate from these estimates rapidly, when the standard deviation increases even only by a small amount. Interestingly, even when the kurtosis is very close to the Gaussian, there is a mismatch in the proportion of positives at each cut-off value. The more conservative the cut-off value (i.e. 2.5σ or higher), the larger the inflation in the number of positives observed when data are lognormal. Even when the shape of the lognormal distribution looks almost indistinguishable from the Gaussian, there is an increase of 1156% in the number of positives.

These results show clearly that even apparently harmless small deviations from the skewness and kurtosis of a Gaussian can lead to systematic misinterpretation of test results and a dramatic increase in the number of false positives. Apparently, no deviation from the Gaussian distribution is too small to have no severe consequences for diagnostics. Since the distribution parameters we employed in the present study come from published qEEG databases, one can be certain that the inflation of false positives occurs on a daily basis, as it is practiced worldwide and fulfills the prerequisites to be considered at least as Questionable Research Practice [19]. The only way to circumvent this problem is to ascertain that the qEEG data basis has exactly the desired properties and can be trusted to produce an acceptable number of false-positives. For too long, the necessity to enforce higher standards for the commercially availability of qEEG databases has been neglected and the relevance of deviant results downplayed. One possible reason for that is the uncritical willingness to believe in the power of "neurotechnology", the so called "neuroenchantment" [20], that affects both laymen and specialists.

Skewness and kurtosis alone are not sufficient to characterize data distributions in general. The reason for that is straightforward: Skewness and kurtosis offer only a global description of

distribution parameters and do not reveal all the measurement properties necessary to define cut-off values for diagnostic purposes. Moreover, as shown by our Simulations 1 and 2, popular cut-off values for the interpretation of skewness and kurtosis values [21, 22] are not appropriate for the construction of test norms. As shown in the present study, skewness and kurtosis only deviated modestly from the expected values of a Gaussian distribution. Nonetheless, these deviations were sufficient to engender severe discrepancies between the density functions and inflation in the number of positives. Accordingly, even slight deviations from the Gaussian distribution led to a dramatic increase in the number of observations with values more extreme than the usual cut-off values. Inflation of the number of individuals with the diagnosis label of "abnormal" EEG increases dramatically with the distance between the cut-off value and the mean. While a cut-off value of 2 standard deviations leads to an inflation of 32% to 96%, a cut-off value of 3 standard deviations leads to an inflation of over 1100% in the number of individuals with an EEG classified as "abnormal". The results of the numerical simulation are in line with empirical findings of a large number of false positive qEEG results (see [23] for a review). If our simulations are accurate, in the clinical setting, such cut-off values misguide the majority of all diagnostic recommendations. Depending on the EEG parameter, the number of false positives is much larger than the number of correct positives, meaning that interventions such as neurofeedback may have been futile in the majority of the studies using qEEG hitherto. Therefore, it is crucial for companies offering qEEG services to provide sufficient evidence that the data they employ to calculate norms do follow a Gaussian distribution, if cut-off values of 2 or even 3 standard deviations are going to be employed in the future. Since evidence of normality of these data has barely been presented in the past, the general validity of research and clinical diagnostics based on qEEG is more than questionable.

Not only the initial diagnostics by means of neurometry seems to be problematic, but also the evaluation of intervention outcomes. As revealed by Simulation 2, heteroscedasticity leads to a considerable inflation of the number of cases showing "spontaneous" improvement in qEEG parameters. Not only the probability of having the EEG classified as "abnormal" is highly inflated, but also the probability that by a second evaluation the EEG will be classified as "normalized", in both cases by force of randomness alone. Here, test-retest reliability estimates lead to false confidence in the usefulness of non-Gaussian qEEG for individual diagnostics. As revealed by Simulation 2, the probability of observing large fluctuations in qEEG scores, when the first measurement was far from the average, is highly inflated, particularly when test-retest reliability is excellent. These results seem to contradict common wisdom on the impact of test-retest reliability on the stability of test scores. The intuition of contradiction is correct in case of a homoscedastic Gaussian distribution, but not valid under heteroscedasticity, particularly when the measurement error is much larger at the tail. Exactly this was shown by Simulation 2. When considering the outcomes of both Simulation 1 and 2 together, a pattern of generation of false positives emerges, which implies for the client, first, an expectation of need for an intervention and then the conviction of benefit from that intervention.

Our results are not completely new, since [24] already observed a high number of false positive results when analyzing EEG data from typical participants. Those early results fit the results of the present stimulation, but did not get any attention in the meantime. Up to now, the study by [24] was cited only 16 times according to google Scholar.

The consequences of this for the clinical practice of neurometry are manifold: most of the diagnostic decisions based on the popular cut-off criteria as performed hitherto are probably wrong, with ethical and legal implications for the therapeutic use of qEEG. Comparisons of skewness and kurtosis as performed in the past are insufficient to guarantee that qEEG norms are useful for clinical and scientific applications. A much more detailed approach is necessary to ascertain the psychometric properties of data, particularly regarding the properties of the

tails of data distributions, since these are particularly relevant for diagnostic purposes. Since lognormal distributions may converge slowly, authors are ethically obligated to present much more robust arguments that the sample estimates of their databases are stable and close enough to population values. In other words, qEEG based diagnostic decisions of "abnormal EEG" as performed hitherto based on cut-off values of 2 or 3 standard deviations has a good chance of being dramatically inflated -and therefore wrong- in a large proportion of the cases. To dismiss such concerns regarding the legitimacy of qEEG, a much better description of the properties of the distribution of normative data is necessary and a more adequate way to account for the instabilities observed at the tails of the distribution should be taken seriously.

An inflation of the number of false positive diagnoses of abnormal QEEG has both material and immaterial costs for its users. Material costs can be substantial, for neurofeedback treatment typically takes several sessions to be completed (see tables in [25] for a summary of treat duration), requires the utilization of specialized equipment and time of an expert neurofeedback trainer. In a recent study, [26] found out that about 73% of users paid for neurofeedback out-of-pocket. As observed by [26], about 80% of neurofeedback users use it for indications not adequately supported by scientific evidence although more than half (57.6%) considered neurofeedback to be a scientifically well-established therapy. This disconnection between scientific evidence and experiences of users can be considered part of the immaterial costs of neurofeedback. These immaterial costs may be even more substantial for the individuals with a false diagnose of "abnormal" EEG. These persons are confronted with concerns regarding their health and cognitive capabilities. False positive diagnosis is known to induce psychosocial harm in affected individuals [27, 28]. Although not as grave as a diagnostic of psychiatric disorder, some form of psychosocial harm can be expected from false-positive EEG diagnosis as well.

As we showed in Simulation 2, one consequence of a skewed distribution of EEG is the volatility of more extreme observations, which is much higher than the test-retest reliability would predict, so that typical individuals also produce extreme EEG with a relatively large probability. However, beyond the problems of a psychometric approach to the quantification of EEG we pointed out in the present manuscript, it is still a legitimate question, whether the fat tailedness anyhow reflects real clinical abnormality and how they are related to clinical conditions. Present evidence seems to reveal no clear association between psychiatric disorders and alterations in EEG frequency bands [29] beyond a trend for elevated power in lower frequency bands and reduced power in higher frequency bands in ADHD, schizophrenia and OCD. The meta-analysis of 184 EEG studies by [29] compared several psychiatric populations and controls and only revealed small effects and low correlations between EEG and symptom severity. In practical terms, these results support the view that the usefulness of EEG as a marker to evaluate the effect of interventions is in the best case very limited. It is possible that a more detailed analysis of the spatiotemporal features of EEG with the help of deep-learning or other sophisticated mathematical modelling techniques will disclose patterns of abnormal EEG with a higher degree of specificity than traditional neurometry is able to. This is however not yet reality but just speculation. Anyways, understanding tail properties of distributions is usually laborious and mathematically not trivial, requiring a considerable quantity of data which probably cannot be achieved with a few minutes of EEG recording (or less!) as frequently argued in the neurometry literature.

## Recommendations

Describe extensively the distribution of qEEG parameters. Thoroughly describing the properties of higher moments of the distribution of normative EEG data is essential to guarantee that

the diagnostic process is transparent and the number of false-positives is reduced to more reasonable levels. Conventional tests such as the Shapiro-Wilks test may not be sufficient to describe fat-tailedness [30]. In line with the recommendations by [31] to discourage Questionable Research Practice, providers of qEEG databases should disclose in detail all the properties of the data they contain. This means reporting skewness, kurtosis as well as confidence intervals for every parameter listed in qEEG databases. QQ-Plots should always be presented when databases are published or services sold.

Cut-off values based on the percentiles of the EEG distribution could be an alternative. Depending on the properties of qEEG data distributions, percentile values could be used instead of the standard deviations. On the one side, percentiles force the developers of databases to put more effort in data collection when creating databases stratified for age, schooling, and sex of participants, since only in large samples it is possible to uniquely determine the 95th and 99th percentiles that are so important for practical purposes. On the other hand, the percentile values are as intuitive as standard deviations and have a straightforward meaning even for members of the family of fat-tailed distributions. However, the usability of percentile values can be limited by heteroscedasticity. If the values in the tails show larger variability than those close to the center, the meaning of extreme percentile values may also be limited, since the true value of that observation is accompanied by a large interval of uncertainty.

Log-transformations are not necessarily sufficient. Some distributions cannot be bent to the shape of a Gaussian by the logarithm [9]. As shown by [10], the distribution of EEG parameters may deviate from Gaussian in ways more extreme than logarithmic compression. In these cases, the logarithm may reduce the degree of fat-tailedness, but as our results clearly show, the remaining fat-tailedness can still be sufficient to inflate considerably the number of (false) positives. Therefore, one should ideally always determine the transformation capable of adequately bending the shape of the distribution of qEEG data into the Gaussian. One practical way to do that is to use other families of transformations [10, 32, 33]. However, this step of data processing is not trivial, since estimating distribution moments and properties when distributions are more pronouncedly fat-tailed requires much more data than the usual few hundreds or thousands available in qEEG data bases. Since the transformations assume that available data are representative of the population distribution parameters, large misestimation problems are still likely.

Include qEEG norm construction in the CRED check-list for the quality of neurofeedback studies [29]. One should take the matter of designing norms for EEG data much more seriously than hitherto. When disclosing the real sample properties of qEEG databases, it is possible that most studies of the past have employed too liberal criteria to define clinical samples. This can produce gross distortions in correlations between treatment outcomes and other outcome measures, distortions in the response to treatments, etc. All these factors decrease the quality of neurofeedback services, lead to useless and expensive treatments and jeopardize reproducibility of study results. For all these reasons, we believe that good quality qEEG norm parameters have to be checked during neurofeedback studies. The "Consensus on the reporting and experimental design of clinical and cognitive-behavioral neurofeedback studies" (CRED-nf checklist) contains a checklist for registering and publication of NF studies [34]. Here we argue that in future versions of the CRED the use of safe neurometry instruments should be included as a positive point and the use of problematic databases a negative point to be included in the characterization of the quality of neurofeedback studies.

A new business model and more training are necessary to improve qEEG guided practices. Since accuracy of qEEG based diagnostics depends on the higher moments of the distribution of qEEG parameters, these have to be thoroughly disclosed in much more detail than is usually the case to allow clients an informed choice. Users of qEEG databases have to receive better

training on when and under which circumstances qEEG is relevant for diagnostic decisions and the evaluation of interventions.

## Limitations

One important limitation of the present study is the limited availability of information on the distribution of qEEG data available in the literature. With more precise and transparent numerical values, the estimates we can provide will become more accurate. Still, our main results as well as their alarming consequences for the practice of qEEG would not be challenged, just become some decimal places more precise. We hope to encourage developers of qEEG to be more accurate in describing sample parameters and diagnosing fat-tailedness in their datasets and users to demand this information before employing qEEG in research and diagnostics in the future.

## Supporting information

**S1 Fig. t-distribution: We generated samples of n = 1000 observations using the t-distribution with 3, 5, 8, 12, 16, 20, 30 and 40 degrees-of-freedom.** The lower the degrees of freedom of a t-distribution, the more fat-tailed it is. In the S1 Fig the ratio of false-positives(t-distribution)/false-positives(Gaussian) is displayed as a function of the ratio kurtosis(t-distribution)/kurtosis(Gaussian). The larger this last ratio, the more fat-tailed is the distribution.
(TIF)

**S2 Fig. Chi$^2$-distribution: We generated samples of n = 1000 observations using the chi$^2$-distribution with 1, 2, 3, 5, 8, 12, 16, and 20 degrees-of-freedom.** The lower the degrees of freedom of a chi$^2$-distribution, the more fat-tailed it is (Fleishman, 1978, p. 528). In the S2 Fig the ratio of false-positives(chi$^2$-distribution)/false-positives(Gaussian) is displayed as a function of the ratio kurtosis(chi$^2$-distribution)/kurtosis(Gaussian). The larger this last ratio, the more fat-tailed is the distribution.
(TIF)

**S3 Fig. Pareto distribution: We generated samples of n = 1000 observations using the Pareto distribution with a location of 1 and exponents 0.5, 1, 1.5, 2, 2.5, 3, 5, and 8.** The lower the exponent of a Pareto distribution, the more fat-tailed it is. In the S3 Fig the ratio of false-positives(Pareto distribution)/false-positives(Gaussian) is displayed as a function of the exponent of the distribution. The difference in the x-axis to the other plots is due to the fact that the mean, variance, and other moments are finite only if the shape parameter a is sufficiently large. The larger this last ratio, the more fat-tailed is the distribution.
(TIF)

**S1 Data. Script for data generation.**
(R)

## Author Contributions

**Conceptualization:** Guilherme Wood, Jan Willem Koten.

**Formal analysis:** Guilherme Wood, Jan Willem Koten.

**Investigation:** Guilherme Wood.

**Methodology:** Klaus Willmes.

**Supervision:** Klaus Willmes, Silvia Erika Kober.

**Validation:** Silvia Erika Kober.

**Visualization:** Guilherme Wood.

**Writing – original draft:** Guilherme Wood, Silvia Erika Kober.

**Writing – review & editing:** Guilherme Wood, Klaus Willmes, Silvia Erika Kober.

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
