## [Decision Letter · Decision Letter 0]

7 Sep 2023

PONE-D-23-14005Fat tails and the need to disclose distribution parameters of qEEG databasesPLOS ONE

Dear Dr. Wood,

Thank you for submitting your manuscript to PLOS ONE. After careful consideration, we feel that it has merit but does not fully meet PLOS ONE’s publication criteria as it currently stands. Therefore, we invite you to submit a revised version of the manuscript that addresses the points raised during the review process.

We look forward to receiving your revised manuscript.

Kind regards,

Kewei Chen, Ph.D

Academic Editor

PLOS ONE

Journal Requirements:

"This research was not funded by any agency."

Additional Editor Comments:

Since the journal editorial office and the AE have had difficulties locating a second reviewer for such long time, we feel the issue raised in this manuscript is important (especially in the context of proper use of clinical resources and financial support). This is also based on the sole reviewer comments of no major concerns.

In addition to responding to the reviewer's comments, the author must more clearly and more specific discuss clinical consequence of the inflated abnormal qEEG, related to excess use of unneccesary services. any quantitative discussion on the extra costs?

On the other hand, the fat tail reflect real clinical abnormality? How these fat or non-fat tail (the tail itself) related to clinical conditions. This should be discussed more

Reviewers' comments:

Reviewer's Responses to Questions

**Comments to the Author**

1. Is the manuscript technically sound, and do the data support the conclusions?

Reviewer #1: Yes

2. Has the statistical analysis been performed appropriately and rigorously? 

Reviewer #1: Yes

3. Have the authors made all data underlying the findings in their manuscript fully available?

Reviewer #1: Yes

4. Is the manuscript presented in an intelligible fashion and written in standard English?

Reviewer #1: Yes

5. Review Comments to the Author

Reviewer #1: In their manuscript entitled “Fat tails and the need to disclose distribution parameters of qEEG databases”, the authors use simulation to show that if the normal distribution assumption is violated for the neurometry parameters, the current analyses of qEEG data could be problematic. The authors also make some recommendations for practice in this area.

Although the results are not surprising as we already know that treating non-normal distribution as normally distributed will have undesirable consequences, overall, this is a clearly written report with clinic significance.

The authors only simulate log-normal data. It might be more convincing to simulate more data from other (fat-tail) distributions to show the serious limitations when current analyses with normal distribution assumptions are used for qEEG data.

6. PLOS authors have the option to publish the peer review history of their article (what does this mean?). If published, this will include your full peer review and any attached files.

Reviewer #1: No

---

## [Author Response · Author response to Decision Letter 0]

7 Nov 2023

PLease see the file attached containing our response to reviewers.

---

## [Decision Letter · Decision Letter 1]

22 Nov 2023

Fat tails and the need to disclose distribution parameters of qEEG databases

PONE-D-23-14005R1

Dear Dr. Wood,

We’re pleased to inform you that your manuscript has been judged scientifically suitable for publication and will be formally accepted for publication once it meets all outstanding technical requirements.

Kind regards,

Kewei Chen, Ph.D

Academic Editor

PLOS ONE

Additional Editor Comments (optional):

Reviewers' comments:

Reviewer's Responses to Questions

**Comments to the Author**

1. If the authors have adequately addressed your comments raised in a previous round of review and you feel that this manuscript is now acceptable for publication, you may indicate that here to bypass the “Comments to the Author” section, enter your conflict of interest statement in the “Confidential to Editor” section, and submit your "Accept" recommendation.

Reviewer #1: All comments have been addressed

2. Is the manuscript technically sound, and do the data support the conclusions?

Reviewer #1: Yes

3. Has the statistical analysis been performed appropriately and rigorously? 

Reviewer #1: Yes

4. Have the authors made all data underlying the findings in their manuscript fully available?

Reviewer #1: Yes

5. Is the manuscript presented in an intelligible fashion and written in standard English?

Reviewer #1: Yes

6. Review Comments to the Author

Reviewer #1: Thank you for considering my comments and the revising the manuscript accordingly. The manuscript looks better now.

7. PLOS authors have the option to publish the peer review history of their article (what does this mean?). If published, this will include your full peer review and any attached files.

Reviewer #1: **Yes: **Zhongxue Chen

---

## [Editor Report · Acceptance letter]

26 Dec 2023

PONE-D-23-14005R1 

PLOS ONE

Dear Dr. Wood, 

I'm pleased to inform you that your manuscript has been deemed suitable for publication in PLOS ONE. Congratulations! Your manuscript is now being handed over to our production team.

Kind regards, 

on behalf of

Prof. Kewei Chen 

Academic Editor

PLOS ONE